# Understanding PITX2-Dependent Atrial Fibrillation Mechanisms through Computational Models

**DOI:** 10.3390/ijms22147681

**Published:** 2021-07-19

**Authors:** Jieyun Bai, Yaosheng Lu, Yijie Zhu, Huijin Wang, Dechun Yin, Henggui Zhang, Diego Franco, Jichao Zhao

**Affiliations:** 1College of Information Science and Technology, Jinan University, Guangzhou 510632, China; tluys@jnu.edu.cn (Y.L.); zyj1934261010@stu2019.jnu.edu.cn (Y.Z.); 2Auckland Bioengineering Institute, University of Auckland, Auckland 1010, New Zealand; 3Department of Cardiology, First Affiliated Hospital of Harbin Medical University, Harbin 150000, China; yindechun0429@163.com; 4Biological Physics Group, School of Physics & Astronomy, The University of Manchester, Manchester M13 9PL, UK; henggui.zhang@manchester.ac.uk; 5Department of Experimental Biology, University of Jaen, 23071 Jaen, Spain; dfranco@ujaen.es

**Keywords:** cardiac arrhythmias, atrial fibrillation, PITX2, computational model, electrical remodelling, structural remodelling, calcium handling, mRNA, electrophysiology

## Abstract

Atrial fibrillation (AF) is a common arrhythmia. Better prevention and treatment of AF are needed to reduce AF-associated morbidity and mortality. Several major mechanisms cause AF in patients, including genetic predispositions to AF development. Genome-wide association studies have identified a number of genetic variants in association with AF populations, with the strongest hits clustering on chromosome 4q25, close to the gene for the homeobox transcription PITX2. Because of the inherent complexity of the human heart, experimental and basic research is insufficient for understanding the functional impacts of PITX2 variants on AF. Linking PITX2 properties to ion channels, cells, tissues, atriums and the whole heart, computational models provide a supplementary tool for achieving a quantitative understanding of the functional role of PITX2 in remodelling atrial structure and function to predispose to AF. It is hoped that computational approaches incorporating all we know about PITX2-related structural and electrical remodelling would provide better understanding into its proarrhythmic effects leading to development of improved anti-AF therapies. In the present review, we discuss advances in atrial modelling and focus on the mechanistic links between PITX2 and AF. Challenges in applying models for improving patient health are described, as well as a summary of future perspectives.

## 1. Introduction

Atrial fibrillation (AF), characterized by rapid and disorganized electrical activation in the upper chambers of the heart, is the most common cardiac arrhythmia [1]. AF affects 2–3% of the population and its prevalence is rising with age from 2.4% in people 65 years old to 10% in those older than 80 years [2]. Recently, a large number of studies have shown that AF, especially lone AF, has an important genetic component [3]. 

The paired-related homeobox gene (PITX2) has been considered to be a potential gene that may trigger AF risk variants on chromosome 4q25 [4,5] and two SNPs (rs2200733 and rs10033464) in chromosome 4q25 were reported [6]. Although various advanced technologies have been developed, their effectiveness is limited and existing treatment regimens are rarely curative [7,8,9]. This result is partly due to our limited understanding of the mechanisms of AF in the first and second stages (Figure 1A). In the third stage, the development of actionable personalized approaches, which take into account patient-specific profiles and arrhythmia mechanisms, will likely be essential to overcome current challenges in AF management (Figure 1B). Computational modelling and simulation are indispensable in cardiac electrophysiology in the study of complex arrhythmias [10,11,12], such as AF. The multi-scale model of cardiac electrophysiology provides a framework that can integrate experimental and clinical findings [13], and link micro-scale phenomena to emerging behaviors of the entire organ [14]. Computational modeling is now an important part of AF mechanism research, because it can supplement experimental observations and suggest novel mechanisms [15,16,17,18,19,20,21,22,23,24,25,26]. In addition, the entire atrial simulation is currently used to design novel and personalized treatment strategies, thereby contributing to the development of precision medicine in cardiology. In this narrative review, we focus on the latest developments in the atrial model in elucidating the mechanism of PITX2-dependent AF. We summarize experimental studies of the role of PITX2 in cardiogenesis and arrhythmogenesis, advances in atrial modelling, and modelling studies for investigating PITX2-dependent AF mechanisms. Specifically, we summarize the progress in the development of multi-scale AF models, and then focus on the mechanistic connection between alternations in atrial structure and electrophysiology with PITX2-dependent AF from the perspective of computational modeling. We focus on how AF modeling can supplement experimental data in ways that cannot be achieved outside of the simulation framework, and how AF models can reveal novel AF mechanisms. We conclude the review with a summary of the future prospects of the atrial model’s mechanical understanding of AF, towards the goal of understanding patient-specific AF mechanisms that would allow for personalised treatment.

## 2. The Role of PITX2 in Cardiogenesis and Arrhythmogenesis

It is known that PITX2 plays a key role in establishing left-right asymmetry and in heart development. Therefore, it supports the notion that impaired function of PITX2 might underlie AF. In the following sections, we will introduce the role of PITX2 in left-right asymmetry, summarize the regulation effects of PITX2 on gene networking, microRNAs and membrane effector gene and discuss remodelling arising from impaired PITX2 in AF [28].

### 2.1. Pitx2 Promotes Left–Right Asymmetry

PITX2 plays an important role in the morphogenesis of early development [29,30]. So far, three isoforms have been described: PITX2a, 2b, and 2c, among which PITX2c is the main isoform expressed in the heart [31,32]. PITX2 is a key mediator of developmental signaling cascades involving factors such as Lefty, Nodal, Shox2, Nkx2.5, and Tbx3 [33,34,35].

In the early stage of embryogenesis, PITX2 is directly regulated by the nodal-mediated left-right asymmetry pathway, thereby imparting left morphogenesis to several organs in the body [36]. During heart development, PITX2 is essential for left-right asymmetry, differential regulation of left ventricular identity, and ventricular asymmetric remodeling procedures [34]. PITX2 also confers left atrial morphology [37]. The left atrium with a null mutation of PITX2 has morphological features on the right side, including venous valves and trabecular myocardium [37]. In addition, PITX2-deficient embryos also have bilateral or ectopic sinus nodes, which may explain the AF susceptibility observed in adult animals with decreased PITX2 expression [38,39,40]. PITX2 also plays a vital role in regulating the development of the cardiac conduction system and pulmonary myocardium [35,39]. 

All the cardiomyocytes of the heart had pacemaker properties at first, but only a small part of the cells differentiated into pacemaker cells, forming the cardiac conduction system, including the sinus node [35]. PITX2 inhibits Shox2, which is a transcription factor expressed in the precursor of sinoatrial node, which may lead to down-regulation of nodule gene programming and up-regulation of Nkx2.5, thereby inducing gene programming of working myocardial phenotype [41]. The upstream Lefty1 restricts PITX2 expression to Shox2 expression, consequently, it only inhibits the development of the sinoatrial node in the left atrium. In the right atrium, the lack of PITX2 will cause Shox2 upregulation, which can prevent Nkx2.5 but induce Tbx3, which is essential for the development of the sinus node (Figure 2) [42]. Moreover, there was a deficiency in the contribution of the PITX2-expressing lineage to pulmonary veins in PITX2c homozygous mutant embryos [39] because of a defect in pulmonary vein precursors [43]. The pulmonary vein is a well-known site that promotes ectopic activity inducing spontaneous AF [12].

### 2.2. Gene Regulatory Mechanisms Driven by PITX2

In addition to Shox2, Nkx2.5 and Tbx3, PITX2 can modulate the expression of Zfhx3, Il6r, Cav1, Syne2, Wnt8a and Tbx5 (Figure 3). Lozano-Velasco et al. used a series of gain and loss-of-function methods to demonstrate that PITX2 can regulate the expression of Zfhx3 and Il6r [44]. The role of Zfhx3 in AF remains to be explored, whereas the involvement of Il6r could be related to the inflammatory process [45] that, if damaged, is related to AF. More recently, Lozano-Velasco et al. demonstrated that PITX2 modulates Cav1, Syne2, Wnt8a [44,46]. Risk variants related to Cav1 and Syne2 have been reported in other cardiac electrophysiological diseases, but their functional role is still unclear. Cav1, a structural protein of caveolae, affects the activity and biogenesis of nitric oxide and regulates signal transduction pathways that mediate inflammation and oxidative stress [47]. Syne2 is a member of a protein family found mainly in the outer nuclear membrane and other subcellular compartments, and plays a role in nuclear migration, nuclear localization during retinal development, and ciliogenesis [48]. Importantly, impaired Wnt8a expression was observed in PITX2 dysfunction models with basic ECG changes but not in those with normal ECG records [44]. Gain- and loss-of-function approaches demonstrated that Wnt8a signaling can modulate the expression of calcium handling proteins, contributing to triggered activity during AF onset. Recently, a transcriptional network between Tbx5 and PITX2 was identified. Tbx5 directly activated PITX2, while Tbx5 and PITX2 antagonize the regulation of membrane effect genes.

### 2.3. microRNAs Regulated by PITX2

In addition to gene regulatory mechanisms driven by PITX2, there are several pieces of evidence supporting that PITX2 regulated microRNAs that contribute to the pathophysiology of AF [50,51]. MicroRNAs involved in the pathophysiology of PITX2-dependent AF include miR-106-25, miR-17-92, miR-29a, miR-200, miR-203, miR-21, miR-208ab, miR-1, miR-26b and miR-106ab [52,53]. Wang et al. demonstrated that miR-17-92 and miR-106b-25 are under the control of PITX2 and genetic deletion of these microRNA clusters are susceptible to pacing-induced AF [54]. More recently, a large number of microRNAs modulated by PITX2 were identified to be associated to AF in patients [44]. In the absence of PITX2, miR-21, miR-106a, miR-203 and miR-208ab were down-regulated, whereas miR-1, miR-26b, miR-29a, miR-106b and miR-200 were up-regulated. In these microRNAs, several microRNAs (including miR-1, miR-133, miR-21, miR-106b and miR-26) have been reported to regulate calcium, sodium and potassium channel subunits [55]. Overall, these data demonstrate a highly complex gene regulatory network leading to PITX2-dependent AF as summarized in Figure 3.

### 2.4. Membrane Effector Genes Regulated by PITX2

Several potassium channel, sodium channel and calcium handling genes are regulated by PITX2, as seen in mutant PITX2 models. TASK-like background currents, contributors to the resting membrane potential, decreased in PITX2c^+/−^ mice [56]. In the left atrial chambers, Kir2.1 (Kcnj2) and Nav1.5 (Scn5a) channel expression are decreased [57], while Bmp10 is highly upregulated [58]. These findings support that PITX2 may control atrial chamber dimensions, because overexpression of Bmp10 plays a crucial role in regulating physiological hypertrophy. In the right atrial myocytes, overexpression of PITX2c increased *I_Ks_* density and reduced *I_CaL_* density [59]. In additional to changes in ion channels, impaired Gja5 expression was found in adult heterozygous PITX2 mutant mice [42,57,60]. Moreover, several calcium handling genes were found to be regulated by PITX2. In atrial-specific NppaCrePitx2 mice, Cacna1c is significantly down-regulated, whereas Atp2a2, Casq2, and Pln display a major and RyR2 a minor up-regulation [44]. Recently, similar defective calcium homeostasis links the 4q25/PITX2 variants to the risk of AF [41,61]. Experimental results showed that human atrial myocytes from patients carrying the PITX2 variant (rs13143308T) display increased Serca2a expression, SR calcium load, and RyR2 phosphorylation, which likely may cause triggered activity observed in these patients [61].

### 2.5. Remodelling Linked Impaired PITX2 to AF

It has been found that both overexpression and inhibition of PITX2 are related to human AF. In patients who need ablation of AF, PITX2 is expressed in a wide range in the entire tissue sample of the left atrial appendage and the gradient of PITX2 mRNA is increased [58]. PITX2 concentrations in whole left atrial tissue were similar in patients with and without AF recurrence. Morphological analysis of the patient’s left atrial appendage tissue biopsy showed that the tissue heterogeneity of some samples had obvious fatty deposits and fibrosis, while other samples had high myocardial content. However, the concentration of PITX2 in left atrial appendage cardiomyocytes of patients with recurrent AF is lower than that of patients without relapse. Therefore, the recurrence of AF is related to the decrease of PITX2 concentration in myocardial cells but not the entire left atrial appendage tissue PITX2. This study supports the hypothesis that decreased PITX2 concentration in left atrial cardiomyocytes is associated with recurrence of AF. Furthermore, patients with chronic AF have a higher concentration of PITX2 in the myocardial cells of the right atrial appendage compared with sinus rhythm [59]. Further analyses showed that the expression of PITX2 was positively and negatively correlated with *I_Ks_* and *I_CaL_* densities, respectively. These results supported that impaired PITX2 in right atrial appendage cardiomyocyte was associated with electrical remodelling, contributing to AF.

Further analysis showed that remodelling (including electrical remodelling, structural remodelling and calcium handling) links impaired PITX2 (upregulated or downregulated PITX2) to AF (Figure 4). Electrical remodelling includes changes in *I_Na_*, *I_K1_*, *I_Ks_* and *I_CaL_*. In NppaCrePitx2-deficient mice, the expression of Scn5a and Scn1b was impaired in both left and right atrial chambers, whereas the expression of Kcnj2, Kcnj12, and Kcnj4 was reduced in the left atrial myocardium. Consistent with these findings, western blot analysis showed that Kir2.1 (Kcnj2) and Nav1.5 (Scn5a) channel expression is decreased in the atrial chambers of NppaCrePitx2-deficient mice. Further function analysis showed that cardiomyocytes of the left atria displayed a depolarized resting membrane potential and a smaller action potential amplitude than those from control mice [57]. However, an increase [60] and no change [56,62] in Scn5a expression were also observed in other studies. In addition to *I_Na_* and *I_K1_*, remodelled *I_Ks_* and *I_CaL_* were associated with PITX2c overexpression in right atrial myocytes [59]. Experimental results showed that the expression of PITX2c was positively and negatively correlated with *I_Ks_* and *I_CaL_* densities, respectively. Interestingly, genetic heritability may be mediated via definable atrial remodeling in chronic AF [63,64].

PITX2-dependent calcium handling may link impaired PITX2 to spontaneous atrial ectopic beats. Experimental results showed that calcium handling is impaired in atrial-specific Pitx2 mutants in a dose-dependent manner [44]. In detail, Cacna1c is significantly down-regulated, whereas Atp2a2, Casq2, and Pln display a major and RyR2 a minor up-regulation. A recent study showed similar defective calcium homeostasis arising from the 4q25/PITX2 variant rs13143308T included increased Serca2a expression, SR calcium load, and RyR2 phosphorylation, which likely cause the abnormally high incidence of both calcium release-induced triggered activity [61].

In addition to calcium handling and electrical remodelling, PITX2-dependent structural remodelling includes alteration in gap junction and its distribution, fibrosis and cell proliferation [66]. The decrease of PITX2 concentration in myocardial cells is related to the recurrence of AF. Compared with patients with no recurrence of AF, patients with recurrent AF have a lower concentration of PITX2 in left atrial appendage myocardial cells. In the patient’s left atrial appendage tissue, obvious fat deposition and fibrosis also were found in some specimens, while high myocardium content was observed in other specimens. Up-regulated Bmp10 linked structural remodelling to imparied PITX2 [58], because the overexpression of Bmp10 plays a key role in regulating physiological hypertrophy [67]. Experimental results also showed that the downregulation of Pitx2 was inversely correlated with the upregulation of miR-21 which may play a vital role in the development of fibrosis by promoting the proliferation of interstitial fibroblasts and increasing the abnormal deposition of the extracellular matrix [68]. Therefore, reduced PITX2 may contribute to the development of fibrosis. PITX2 also regulated the cell-cell gap junction gene (Gja1, Gja5). The expression of Gja1 (Cx43) was highly up-regulated, while the expression of Gja5 (Cx40) was severely down-regulated in the left atrial chamber of the atrial-specific Pitx2 mutant. In addition, Gja1 (but not Gja5) showed a dose-dependent regulation of PITX2 and was upregulated in both left and right atrial chambers [44]. Clinical studies support this notion changes in gap junctions arising from impaired PITX2 may influence electrical conduction. Carriers had relatively increased conduction velocity heterogeneity, complex signals, regional left atrial slowing, or conduction block particularly in the posterior and lateral walls [64].

## 3. Recent Advances in Atrial Modelling

Based on multiple experimental data (such as CT, MRI, LGE-MRI, electrocardiography, genetics, proteins and membrane potential measurements) obtained from patients, multi-scale mechanistic models are developed by using systems of differential equations [69]. For example, models of ion channels across the cell membrane, the intracellular calcium kinetic model, the regulation model of calcium signal, myofilament mechanics model and microstructure model are the important parts of the cell model. Considering cell models, gap junction model and fibroblast-myocyte coupling are integrated into the tissue structure to form a basic tissue model. Base on the basic tissue model, atrial geometry, fiber orientation and fibrosis distribution are used to establish a whole-atria model. Using multi-scale atrial models, the simulation of PITX2-induced remodelling can perfectly control parameters and observe all model components, making it very suitable for testing causality and exploring AF mechanisms [70]. In detail, novel mechanistic insights from multi-scale atrium models would provide prognostic support, therapeutic support and diagnostic support (Figure 5). In the following sections, we will introduce recent advances in models of ion channel, calcium handling, atrial cell and atrial geometry.

### 3.1. Ion Channel Modelling

Ion channels, specialized proteins in the plasma membrane, provide a passageway through which ions can move downward along the electrochemical gradient of the plasma membrane. The movement of ions can generate an ion current which is determined by the driving force, dynamic opening probability of ion channels and the maximum conductance [72]. Ion-channel open probability can be modelling with three common approaches (i.e., instantaneous and time-independent functions, Hodgkin–Huxley representations and Markov formulations) for two types ion channels. For ionic currents such as *I_K1_*, the open probability is usually represented as an instantaneous state variable [73,74]. For other currents such as the sodium current with several gating variables operating independently or with a number of dependent states, their open probabilities can be represented as Hodgkin–Huxley functions, or Markov models [75] (Figure 6). In Hodgkin–Huxley representations, each kinetic characteristic is represented with a gating variable, which dynamically changes according to “open” and “close” rates, which depend on driving forces (e.g., membrane potential) [11]. All kinetic properties in the process independently operate and thereby the open probability is calculated by multiplying the individual gating variable. In Markov formulations, the movement of charged ions through membrane channels is represented by a number of dependent states (e.g., closed type 1 (C1), closed type 2 (C2), open (O) and inactivated (IS)). Each dependent state variable reflects the fraction of ion channels currently residing and state transition rates can be depended on any state variable. It is worth noting that each Hodgkin–Huxley model has a Markov equivalent, but the converse is not the case because Markov models are not limited to the same assumptions (identical and independent gating particles). Within ion channel modelling, the kinetic parameters need to be estimated. For any two-gate Hodgkin–Huxley model, there are four ways to fit a voltage-dependent ion-channel model to whole-cell current experiments [76]. The first method is to use the voltage clamp protocol to measure the ion current and analyze it to obtain the time constant and steady state of multiple voltages. Curves are fitted to the obtained points and the estimated parameters are inserted into the ion channel model. Different from the first way, the second way conducts simulations to obtain simulated steady states and time constants. The difference between the experimental and simulated data points is calculated and minimized to determine the estimated parameters using numerical optimization. Like the second way, the third way considers the differences between the experimental and simulated data points, but these data points are from the current traces instead of the summary curves. Like the third method, the fourth method uses a single compressed voltage clamp protocol instead of a series of classic protocols to take advantage of the difference between experimental and simulated current traces. The advantage of the first and second methods is that only steady-state and time constants are required as inputs, which are easy to find in the experimental literature.

### 3.2. Computational Modelling of Atrial Cell

The ions flowing through the ion channels in the plasma membrane generate currents, and these currents cause a characteristic change in the membrane voltage, called the action potential. This action potential triggers the release of calcium from the internal stores (i.e., the sarcoplasmic reticulum) through a process called calcium-induced calcium release, which results in an instantaneous increase in the internal calcium concentration. These calcium ions interact with myofilaments to trigger cell contraction, but also impact diverse signaling cascades and influence the regulation of gene expression. In order to capture these changes of atrial cells, atrial cellular electrophysiology, calcium handling, calcium signaling and myofilament contraction should be considered in the atrial cell models (Figure 7). Computational modelling of components (including cellular electrophysiology [72,77,78], calcium handling [79], calcium signaling [80,81] and myofilament contraction [82]) has been previously reviewed. Here, we mainly summarize the existing human atrial cell models.

Based on human cellular electrophysiology data, the first two computational models (Courtemanche and Nygren models) were developed [83,84]. Hodgkin–Huxley/instantaneous ion-channel formulations for describing the kinetic processes and common-pool models for approximating the cytosol as a homogeneous compartment without considering local changes in intracellular ion concentrations were used in both models. However, the long-term stability of the Nygren model hinders its wide application. Its stability was improved by incorporating charge conservation for the stimulus current [85] and action potential rate dependence was subsequently updated by modifying formulations of potassium currents [86]. Based on the modified Nygren model [86], Koivumäki et al. divided the cytosol and sarcoplasmic reticulum into several horizontal components with centripetal calcium diffusion between them and developed the first model with a partial spatial representation of the atrial cardiomyocyte [87,88]. Along with the Koivumäki model, Colman et al. extended the Courtemanche model by incorporating new formulations for *I_to_* and *I_Kur_* and for intracellular calcium handling [89]. In additional to several adaptations of Courtemanche and Nygren models, Grandi et al. [90] and Bai et al. [91] made changes to human ventricular models to generate the human atrial models. In details, based on new experimental data, the Grandi model was modified to include sodium-dependent regulation of *I_K1_* and *I_K,Ach_* [92] or incorporate the two-pore potassium current and its regulation [93]. These models do not simulate local control of calcium processing and assume that all parts of the cell have the same calcium processing behavior. They do not include studies of calcium sparks, sarcoplasmic reticulum calcium-release events, or the subcellular distribution of calcium processing proteins in subcellular regions. Therefore, computational models with detailed spatial calcium processing features have been emerging. For simulations of centripetal calcium waves, Koivumäki et al. developed a spatial calcium-handling model with the 1-dimensional distinct transverse compartments [87,88], whereas the spatial calcium-handling model developed by Voigt et al. is with transverse and longitudinal compartments [94]. Sutanto et al. further modified the Voigt model with the 2-dimensional compartments to simulate calcium waves of axial tubules [95]. Recently, a cuboidal 3D spatial model with various arrangement of subcellular compartments was developed to describe calcium waves more realistically [96].

### 3.3. Geometric and Image-Based Atrial Modeling

Atrial muscle is a composite tissue at the microscopic level. The tissue is composed of atrial cardiomyocytes and fibroblasts supported by extracellular matrix and infiltrated by liquid [97]. Atrial cardiomyocytes are coupled to other cardiomyocytes or/and fibroblasts through gap junctions, so that both cell-to-cell signal transduction and action potential propagation can be achieved (Figure 8). Propagation of the action potential in the atrial tissue is usually modeled using spatially continuous models that are seen as a local spatial homogenization of the behavior of membrane, intra- and extracellular spaces. The current in the tissue structure is usually controlled by using a conductivity tensor field through a single domain reaction-diffusion partial differential equation to the volume of the tissue or organ. Atrial tissue micro-structure (including spatial organization and cellular components of myocardium) and electrical properties have been reviewed extensively elsewhere [98]. In this section we focus on geometric and image-based atrial modeling.

In order to study human AF mechanisms and treatment, a lot of effort has been invested to integrate detailed anatomical, structural and electrophysiological information in the three-dimensional (3D) computer atrial modeling [100,101]. Based on experimental/clinical data on medical imaging and invasively acquired electroanatomic maps, atrial geometry with wall thickness [102], fibrosis distribution [103,104], myofibre orientation, regional electrical heterogeneities and AF driver distribution [105] were used to develop patient-specific 3D models [106]. In details, models with real atrial geometry are reconstructed from medical imaging, specifically from cardiac MRI and/or cardiac CT scans using image segmentation and 3D reconstruction algorithms [107,108,109,110]. In the 3D atria, fibrosis can be detected on late gadolinium enhancement MRI (LGE-MRI) using different thresholding techniques [111,112]. In additional to fibrosis distribution, the orientation of muscle fibers, which has effect on the electrical conductivity, should be incorporated in 3D atrial models using fibre orientation atlases derived from histology, rule-based methods or methods that use morphological data of the endo- and epicardial surfaces and the local solutions of Laplace’s equations [113,114]. Besides these structural heterogeneities (wall thickness, fibrosis distribution and myofibre orientation) [115], electrical heterogeneity also plays important roles in the genesis of AF [13,89,116]. Electrical heterogeneities are mainly characterized by regional differences in action potential morphology and conduction velocity arising from regional variations of ion current and connexin expression [13,89,116]. The regional differences in action potential morphology have typically been incorporated by changing the maximum conductance and gating variables of ion channel models, and differences in conduction velocity can be represented in models by spatially changing the conductivity of the tissue according to the tensor vector obtained from the fiber direction information [117]. The 3D virtual human atria not only allows to differentiate the relative contribution of each variable (i.e., gene variants, ion channels, calcium handling proteins or structural features) to AF mechanisms, but also allows for the development of personalized treatments (e.g., targeted ablation planning and antiarrhythmic drug selection) [10] (Figure 9).

## 4. Mechanistic Insights into PITX2-Dependent AF Using Computational Models

Atrial computational modelling allows integration of detailed basic information about atrial electrophysiology at the cellular and tissue level and relevant simulations provides insight in the fundamental mechanisms involved in initiation and perpetuation of PITX2-dependent AF. Increased triggered activity is the primary mechanism of PITX2-dependent AF initiation. Maintenance of PITX2-dependent AF is associated with remodelling (including electrical and structural remodelling) of the atria. Regions of structural or functional inhomogeneities arising from left–right asymmetry of Pitx2 expression may promote maintenance of re-entrant drivers. Experimental data on PITX2-dependent remodelling have been incorporated in atrial models to predict arrhythmic behaviour, linking the impaired PITX2 to AF with the altered electroanatomical substrate. Key structural and functional alternations that are mechanistically linked to PITX2-dependent AF and have been studied using computational models are calcium handling, PITX2 mutation, electrical remodelling, electrical heterogeneity, APD restitution and atrial wall thickness heterogeneity (Figure 10). Although pulmonary vein (PV) ectopy [119], atrial adipose tissue infiltration, development of repolarisation alternans, presence of atrial fibrosis and its distribution, fibroblast-myocyte uncoupling, and atrial stretch with mechano-electrical feedback have a significant role in the maintenance of re-entrant drivers, current atrial modelling studies have not incorporated these alterations.

### 4.1. Calcium Handling Abnormalities and PITX2-Dependent AF

Atrial modelling studies strongly support that calcium handling abnormalities modulated focal ectopic activity and reentry, which are two major mechanisms control the generation and maintenance of AF. Focal ectopic activity can be triggered through early, delayed and spontaneous afterdepolarizations, whereas reentry can be initiated in the setting of unidirectional block when an activation wavefront propagates around anatomical or functional obstacles and re-excites the site of origin. Under the PITX2-dependent AF, downregulated *I_CaL_* can reduce effective refractory period and thereby wavelength, creating a vulnerable substrate in which reentry can be maintained [19]. Besides reentry, delayed afterdepolarizations due to spontaneous diastolic calcium-release events are associated with PITX2-dependent calcium-handling abnormalities [17,91]. On the one hand, upregulated RyR2 and/or phosphorylation of RyR2 can increase RyR2 current density and/or open probability, leading to inappropriate diastolic SR calcium release and delayed afterdepolarizations. On the other hand, delayed afterdepolarizations can also be resulted from store overload-induced calcium release. Store overload-induced calcium release is caused by excessive calcium influx from increased Serca2a activity. Increased Serca2a activity is associated with upregulated Serca2a or/and phosphorylation of Pln arising from impaired PITX2. Therefore, calcium handling abnormalities due to impaired PITX2 contribute to action potential shortening and delayed afterdepolarizations, promoting the initiation and maintenance of AF.

### 4.2. Electrical Remodelling and PITX2-Dependent AF

Atrial modelling studies on AF have demonstrated that action potential shortening contributes to PITX2-induced AF [19,21,22,91]. Consistent with experimental data on PITX2 insufficiency-induced electrical remodelling, remodelled targets include *I_K1_*, *I_Na_*, *I_CaL_, I_Ks_* and gap junctions. In these remodelled targets, we observed that downregulated *I_K1_* resulted in a more positive resting membrane potential and a prolonged action potential duration, upregulated *I_Na_* contributed to a high overshoot and the maximum depolarization rate, the *I_Ks_* increase and (or) the *I_CaL_* decrease led to a reduction in action potential duration and reduced gap junctions caused slow conduction [21]. The genesis of PITX2-dependent AF may be attributable to action potential duration shortening due to changes in *I_Ks_* and *I_CaL_* and slow conduction resulted from downregulated gap junctions. The combined impact of electrical remodeling on initiation and maintenance of atrial fibrillation can be characterized by atrial tissue’s vulnerabilities to re-entry. On the one hand, PITX2-dependent electrical remodelling increased the spatial vulnerability of atrial tissue by decreasing the substrate size and wavelength due to abbreviated the effective refractory period and reduced conduction velocity. On the other hand, the temporal vulnerability of atrial tissue is measured as the vulnerable window during which unidirectional conduction block and re-entry can be induced by a test stimulus. And the vulnerable window was significantly increased for atrial tissue incorporating remodeling. Consequently, the increase of temporal and spatial vulnerabilities of atrial tissues may explain why the PITX2-dependent electrical remodelling promotes initiation of re-entry.

In addition to AF initiation, the PITX2-dependent electrical remodelling promoted the maintenance of reentry by flattening the action potential duration restitution curves [22]. In the case of the PITX2-dependent electrical remodelling, the maximum slope of the action potential duration restitution curve was reduced. This led to a gradual decrease in action potential duration alternans toward a steady state level and thereby a stable spiral wave at the tissue level. The stable spiral wave was characterized by decreased tip meander area and prolonged life span.

### 4.3. Electrical Heterogeneity and PITX2-Dependent AF

Increased electrical heterogeneity between right atrium (RA) and left atrium (LA) due to down-regulated PITX2 expression implicated in the initiation and maintenance of re-entrant arrhythmias [21]. Since the ratio of PITX2 between LA and RA is 100:1 in the human atria, the extent of remodelling after PITX2 deletion is dependent on the amount of PITX2. Increased electrical heterogeneity may result from the difference in PITX2-induced remodelling between LA and RA. In our simulation study, we observed that electrical remodeling induced by PITX2 deficiency causes action potential duration abbreviation and ectopic depolarizations in LA myocytes, but not in RA myocytes. Thus, PITX2 insufficiency can cause an elevated difference in electrical properties between RA and LA cells, increasing repolarization dispersion in tissue and thereby susceptibility to the development of re-entry. Furthermore, Pitx2 deficiency can also cause LA structural remodeling by regulating cardiac structural genes, increasing electrical and structural heterogeneity between the two atrial chambers. The presence of increased fibrosis and decreased cell-to-cell coupling under structural remodelling further facilitates slow conduction, wavelength abbreviation, triggered activity and the initiation of re-entrant drivers, increasing susceptibility to AF. It is known that PITX2-dependent network regulates cardiac structural genes (Gja1, Gja5 and Dsp) and PITX2-induced structural remodeling leads to fibrosis and cell-cell uncoupling. Also, structural remodeling due to the left-sided PITX2 expression increases the intrinsic heterogeneity (i.e., conduction velocity and wavelength) between RA and LA, facilitating the development of re-entry.

### 4.4. PITX2 Mutation and Familial AF

Functional analysis has shown that PITX2 mutations were linked to atrial fibrillation. The impact of rs138163892 and rs13143308T of PITX2 was investigated using computational models. Mechakra et al. identified a non-synonymous mutation c.619A > G (p.Met207Val, rs138163892) of PITX2 [120]. Functional analysis of the transactivation activity of wild-type and variant PITX2c revealed a gain-of-function of PITX2c (the PITX2 c isoform), leading to an increase in the mRNA level of Kcnh2 (the α subunit of *I_Kr_*), Kcnq1 (the α subunit of *I_Ks_*), Scn1b (the β1 subunit of sodium channels that modulates *I_NaL_*), Gja5 (Cx40), and Gja1 (Cx43). Based on these experimental data, the potential impact of the PITX2c p.Met207Val mutation on atrial electrical activity was investigated using multiscale computational models. Simulated results suggested that electrical and structural remodeling arising from the PITX2c p.Met207Val mutation may increase atrial susceptibility to arrhythmia due to shortened action potential duration, reduced conduction velocity and increased tissue vulnerability, which, in combination, facilitate initiation and maintenance of re-entrant excitation waves [22]. In addition, Herraiz-Martínez et al. identified the 4q25 variant rs13143308T of PITX2. Atrial myocytes from carriers of the rs13143308T variant had a significantly higher density of calcium sparks, frequency of transient inward currents and incidence of spontaneous membrane depolarizations [61]. These alterations were linked to higher sarcoplasmic reticulum calcium loading, Serca2 expression, and RyR2 phosphorylation at ser2808 but not at ser2814 in patients carrying the rs13143308T risk variant. Further computational analysis revealed that these calcium-mediated triggered activities were mainly linked to the gain of Serca2a function but not the RyR2 dysfunction [17].

### 4.5. Clinical Relevance and Challenges

Clinical observational studies have suggested that common SNPs on chromosome 4q25 associated with AF modulates response to anti-arrhythmic drug therapy in patients [121]. Patients with low levels of PITX2 mRNA and AF have also been shown to have improved effectiveness of Class I anti-arrhythmic drug therapy [56]. PITX2 levels vary markedly in human atriums [56,58] and it may be desirable to target AF patients with low PITX2 as a distinct population for therapy. The current limited success of rhythm-control therapy is thought to be in part due to heterogeneity of the underlying substrate, interindividual differences, and our inability to predict response to antiarrhythmic drugs in individual patients [70]. By using mechanistic computational models, these observations of PITX2-dependent effects may help improve rhythm therapy in the future. At present, the relevance of mechanistic computational models is only indirect, serving as a plausibility check for mechanisms proposed based on experimental observations and helping to generate new hypothesis that can subsequently be tested experimentally [15,17,19,21,22,91]. Their recent use in safety pharmacology may further affect clinical practice by guiding the preclinical development of novel antiarrhythmic drugs [16]. Similarly, patient-level cost-effectiveness models may also affect clinical practice by influencing reimbursement policies. The direct clinical application of mechanistic models to guide AF therapy (notably ablation) is emerging, but is currently restricted to a few expert centers. Taken together, currently available models have provided insight into all major components of AF therapy, including antiarrhythmic drugs, ablation and anticoagulation, but their role in the disease management of AF patients is still in its infancy and there are numerous challenges.

(1)Each action potential model has different advantages and disadvantages, with numerous results being model specific.(2)The etiology of AF is diverse, but currently available cardiomyocyte models only have limited options for tailoring models to specific clinical conditions.(3)Only a handful of labs worldwide have the available expertise, computing power and required collaboration between clinicians, scientists and engineers to apply mechanistic whole-atria models in the clinical setting.(4)The extent of personalization of whole-atria models, particularly with regard to electrophysiological properties, remains very limited.(5)Current patient-level models do not incorporate fundamental mechanistic patterns of AF pathophysiology.(6)Integration of mechanistic modeling with “big data” approaches might help to improve AF diagnosis and management.

## 5. Open Questions Regarding Research into PITX2-Dependent AF

(1)A robust method for the identification of the precise spatial distribution of PITX2 throughout human atria is needed. Cardiomyocytes from different atrial locations may exhibit spatial heterogeneities in AP properties reported in humans. The impact of changes in spatial heterogeneities due to impaired PITX2 should be further investigated.(2)Well-designed experiments to assess the PITX2 dependence of changes in electrophysiological properties, such as ionic concentrations and additional ionic currents, could help to re-evaluate anti-arrhythmic drugs that have often been ineffective thus far. Based on these findings, the effectiveness of anti-arrhythmic drugs can be assessed using computational models.(3)There is a need for more accurate PV ectopy models. Future studies should focus on development of accurate models of PV electrophysiology, structure and fibrosis distribution, that can be used to investigate how patient-specific predisposition to PV ectopy, in conjunction with patient-specific substrate, result in the onset and maintenance of PITX2-dependent AF.(4)Adipose-like tissue was found in PITX2 conditional-knockout hearts and morphological analysis of patient left atrial appendage tissue biopsies revealed tissue heterogeneity with marked fatty deposits and fibrosis in some specimens, and high myocardium content in others. Changes in PITX2 -induced remodelling (including adipose tissue deposition, atrial stretch with mechano-electrical feedback, the fibrotic atrial substrate, atrial wall thickness heterogeneity and so on) should be incorporated into the 3D human atria and simulations increase the AF mechanisms due to impaired PITX2.

## 6. Conclusions

It is known that PITX2 plays a key role in establishing left-right asymmetry and plays a key role in heart development. This supports the notion that impaired function of PITX2 might underlie cardiac arrhythmias including AF, but pathophysiological mechanisms linking impaired PITX2 to the risk of AF remain to be explored. This is partly due to a highly complex gene regulatory network driven by PITX2 and the PITX2 downstream pathways involved in pro-arrhythmogenic events. Multiple lines of evidence demonstrate that PITX2 exerts a pivotal role in regulating the expression of distinct ion channels, cell-cell coupling and beta-adrenergic stimulation. Most of these pathways are modulated by microRNAs which are under the control of PITX2 action. The relationship between PITX2 and AF is built with electrical remodelling, structural remodelling and calcium handling abnormalities. However, these experimental and basic research on PITX2-dependent AF is not sufficient for understanding the atrial functional proprieties. Computational models enable synergistic integration of multiple experimental data obtained with the use of different clinical modalities and provide a quantitative understanding of atrial structure and function in PITX2-dependent AF. The advancement of PITX2-dependent AF modeling will continue to be strongly dependent on developments in experimental methodologies, which provide data to constrain, enrich, and validate the models. Of particular importance will be the capability to better resolve the pathophysiological structure of the atria and to fully characterize the complex remodeling in PITX2-dependent AF. Major challenges that lie ahead for computer models of AF include, among others, elucidating the dynamics human AF and detecting rotor locations, as well as understanding the multitude of factors that drive progression of AF in some, but not all, patients.

## Figures and Tables

**Figure 1 ijms-22-07681-f001:**
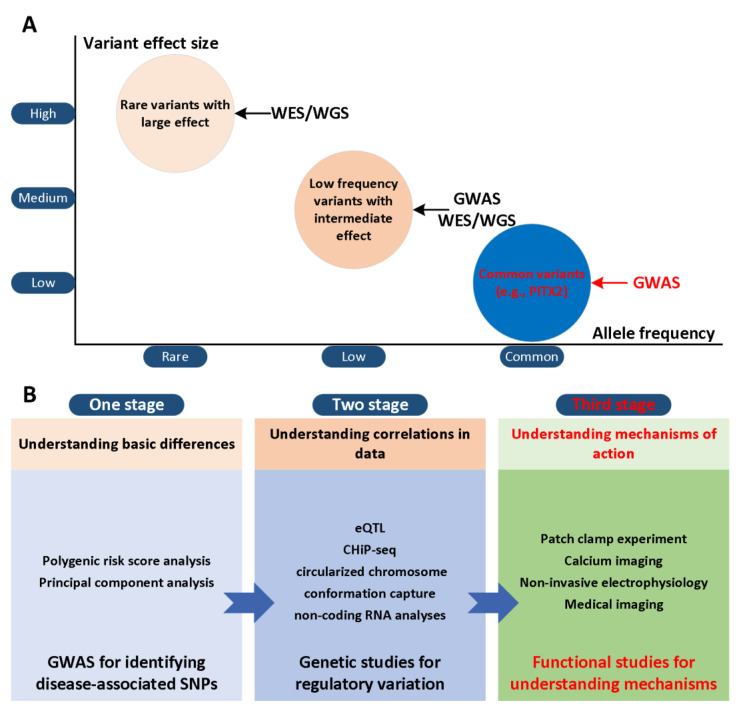
Role of genes in AF from GWAS locus to gene and mechanism. (**A**) GWAS for atrial fibrillation have explored common variants and established many associations with modest to low effect size on disease risk. Whole-exome and -genome sequencing studies seek to further identify low-frequency and rare variants within genes or across the entire genome, respectively. (**B**) The workflow for bridging the genotype-phenotype relationship involves a hierarchical staging of analysis methods: (1) basic gene differences between controls and cases; (2) relevant patterns and correlations in the data [27]; (3) mechanisms of atrial fibrillation in the context of impaired PITX2 that can explain apparent gene differences.

**Figure 2 ijms-22-07681-f002:**
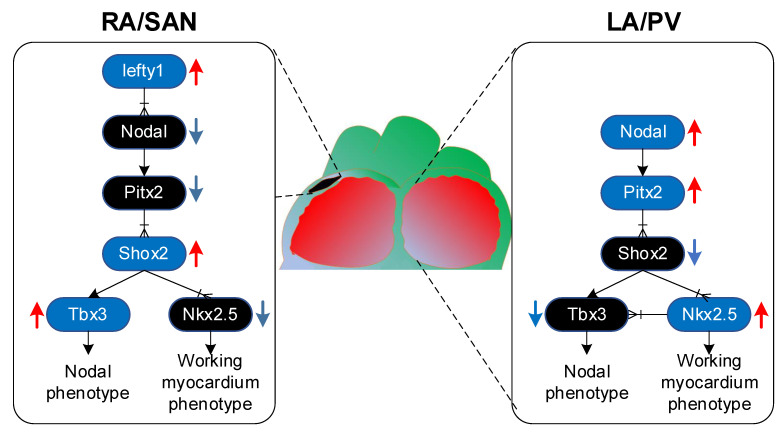
Schematic illustration of PITX2 signaling in atria. Lefty1 represses PITX2 expression in the right atrium (RA), resulting in upregulation of Shox2 that induces sinoatrial nodal (SAN) gene expression through Tbx3. In the left atrium (LA) and pulmonary vein (PV) myocardium, PITX2 is upregulated, leading to downregulation of Shox2 and upregulation of Nkx2.5, which mediates the working myocardium gene expression [31].

**Figure 3 ijms-22-07681-f003:**
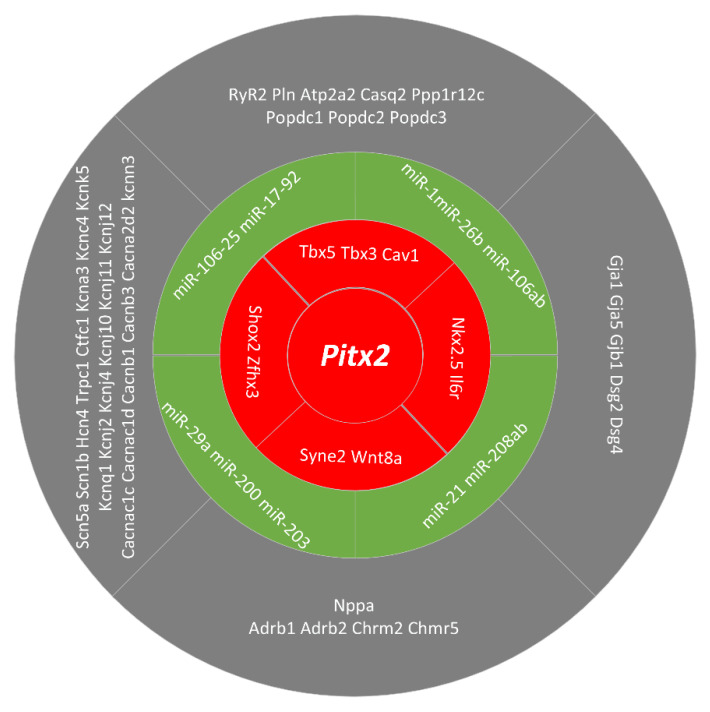
Schematic representation of the PITX2 downstream pathways involved in pro-arrhythmogenic events leading to atrial fibrillation. Multiple lines of evidence demonstrate that PITX2 exerts a pivotal role in regulating the expression of distinct ion channels, cell-cell coupling and beta-adrenergic stimulation. Most of these pathways are modulated by microRNAs which are under the control of PITX2 action [49].

**Figure 4 ijms-22-07681-f004:**
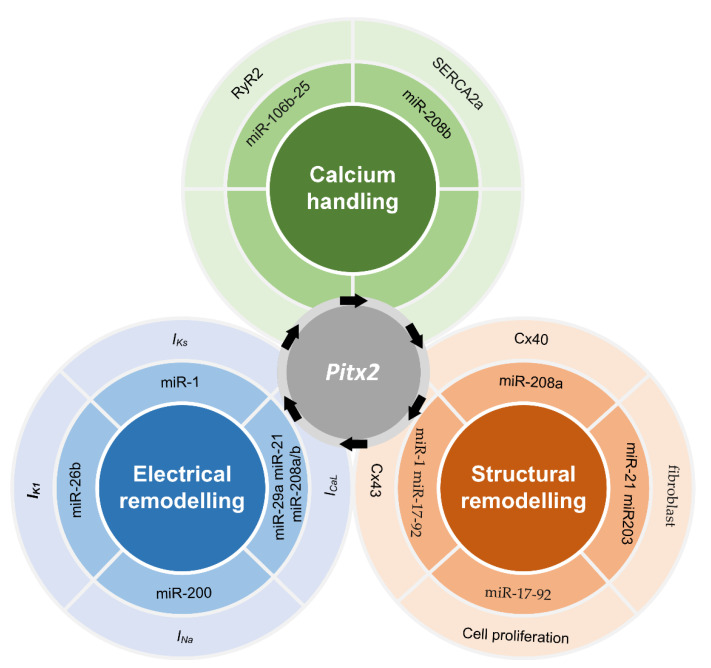
Remodelling linked impaired PITX2 to AF [65]. PITX2-induced remodelling includes electrical remodelling, structural remodelling and calcium handling. miR-26b, miR-200, miR-1, miR-21, miR-29a and miR-208a/b may regulate *I_K1_*, *I_Ks_*, *I_Na_* and *I_CaL_*, contributing to PITX2-induced electrical remodelling. Changes in Cx43, Cx40, fibroblast and cell proliferation due to modulations of miR-208a, miR-21, miR-203, miR-17-92 and miR-1 contribute to PITX2-induced structural remodelling. In addition, abnormalities in calcium handling may arise from the regulation effects of miR-106b-25 and miR-208b on RyR2 and Serca2a, respectively.

**Figure 5 ijms-22-07681-f005:**
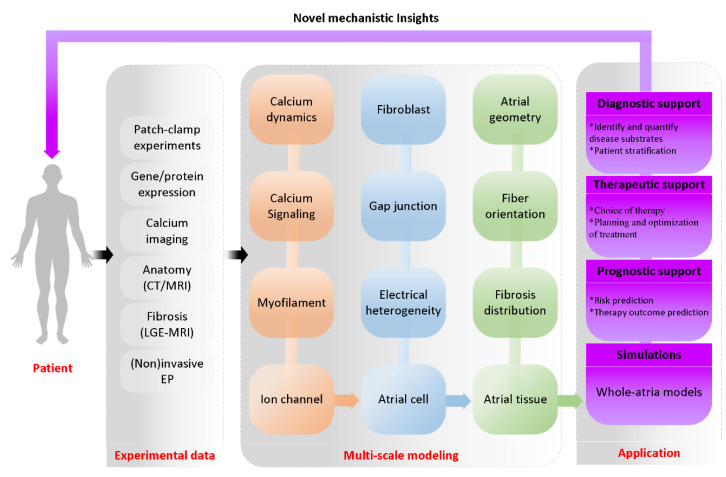
Schematic overview of multi-scale atrial modeling and simulation of 3D virtual human atria. A computational model of the 3D human atria enables synergistic integration of multiple experimental data obtained with the use of different clinical modalities (such as CT, MRI, LGE-MRI, electrocardiography, genetics, proteins and membrane potential measurements) in one personalized heart simulation. The integrative nature of such a virtual-human atria simulation adds value to the existing clinical workflow by offering more quantitative and objective insight into the underlying AF substrates of a patient. In addition, the model provides a platform for virtual evaluation and optimization of a therapy [71].

**Figure 6 ijms-22-07681-f006:**
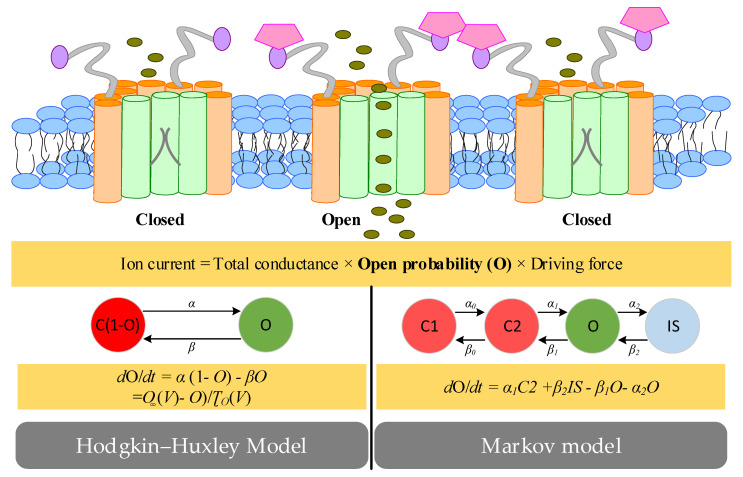
Hodgkin-Huxley-type and Markov-type models of ion channels. Ion currents through specific types of ion channels are determined by a fixed total conductance, the open probability (O) and the driving force. The open probability (O) can be represented as Hodgkin-Huxley functions with a single state variable, or Markov models with a number of states.

**Figure 7 ijms-22-07681-f007:**
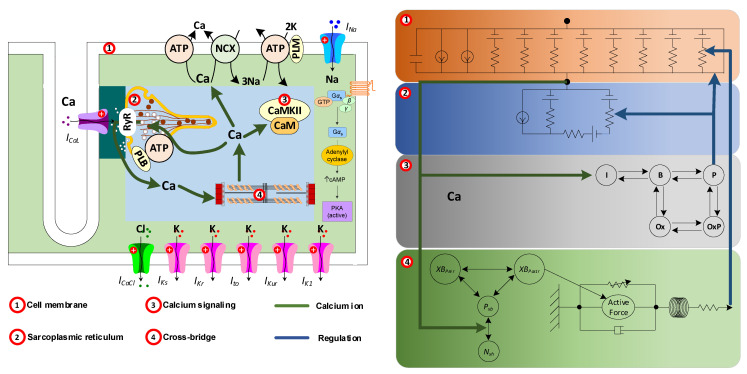
Schematic overview of modelling of atrial cell. A computational model of the human atrial cell roughly includes four modules: cell membrane, calcium cycling, calcium signaling and myofilament. The various modules are mainly connected by calcium ions in the cell and functions (including electrical activity and contractions) of atrial cell are regulated by these modules.

**Figure 8 ijms-22-07681-f008:**
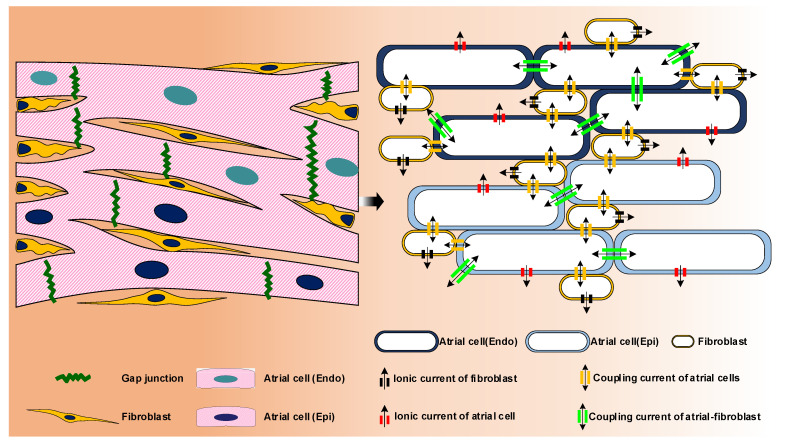
Schematic overview of modelling of atrial tissue. A computational model of the 2D human atrial tissue considered atrial cells, inherent electrical heterogeneity between the endocardial (Endo) and epicardial (Epi) myocytes [99], fibroblast-myocyte coupling, cell-cell coupling via gap junction, the distribution of different cells and fibrosis etc.

**Figure 9 ijms-22-07681-f009:**
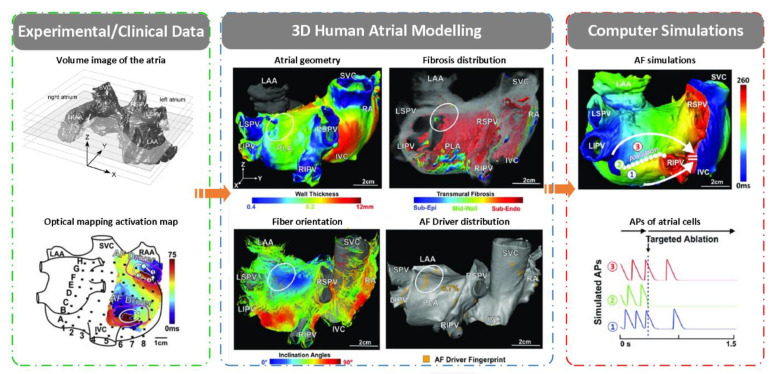
Schematic overview of modeling and simulation of 3D virtual human atria. Based on experimental/clinical data on MRI images and optical mapping activation map [118], atrial geometry, fibrosis distribution, fiber orientation and driver distribution were obtained and were integrated to form a virtual physiological heart considering detailed anatomy. The virtual-human atria simulation offers more quantitative and objective insight in the underlying AF substrates [117].

**Figure 10 ijms-22-07681-f010:**
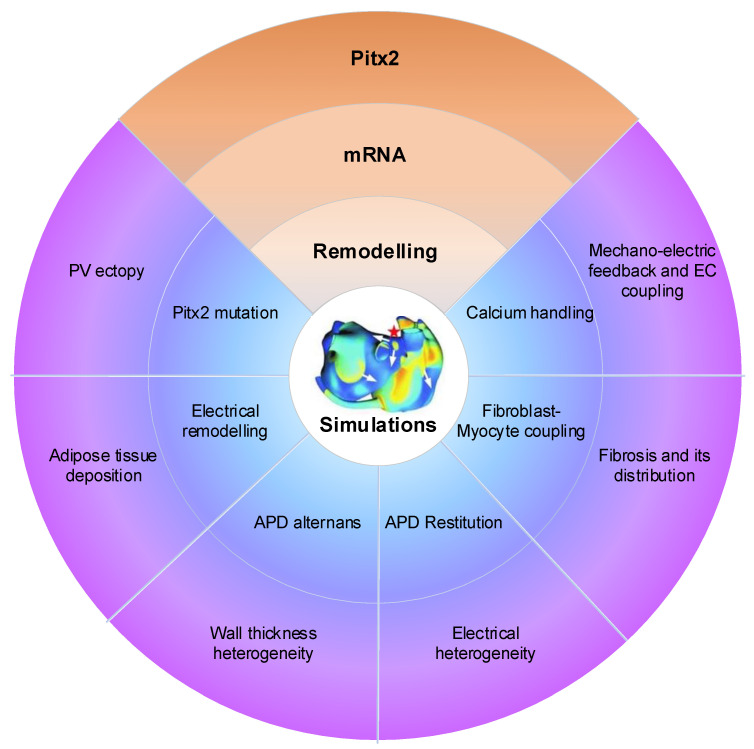
Mechanistic insights from multi-scale AF modeling of remodelling arising from impaired Pitx2. Pitx2 modulates microRNAs and thereby regulates distinct ion channels, cell-cell coupling and beta-adrenergic stimulation, resulting electrical remodelling, structural remodelling and calcium abnormalities. Pitx2-induced remodelling can be incorporated into the 3D human atria and simulations increase the AF mechanisms due to impaired Pitx2. Pitx2 mutation, electrical remodelling, calcium handling, gap junctional uncoupling, development of repolarization alternans, atrial stretch with mechano-electrical feedback, the fibrotic atrial substrate, tissue restitution properties, atrial wall thickness heterogeneity, electrical heterogeneity, adipose tissue deposition and pulmonary vein ectopy.

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
