# Peer review of "Understanding PITX2-Dependent Atrial Fibrillation Mechanisms through Computational Models"

_ijms, 2021, doi:10.3390/ijms22147681_

Round 1
Reviewer 1 Report
Authors have done a good job summarizing PITX2-dependent AF mechanisms. Tables and figures illustrate information clearly. The paper reads well throughout under appropriate headings.
Author Response
Authors have done a good job summarizing PITX2-dependent AF mechanisms. Tables and figures illustrate information clearly. The paper reads well throughout under appropriate headings.
Response: We thank the Reviewer for taking the time to evaluate our study and provide helpful suggestions.
Reviewer 2 Report
The aim of the study was to show PITX-2 dependent atrial fibrillation mechanisms through computational models. The study is scientifically sound. However, I have the following concerns:
- Please adhere strictly to the aim of the study in the Introduction section, which is too general in my opinion.
- Could you please describe the practical implications of the study?
- Could you please show the limitations of the study?
- What is the time frame and type of the included in this review studies?
Author Response
The aim of the study was to show PITX-2 dependent atrial fibrillation mechanisms through computational models. The study is scientifically sound. However, I have the following concerns:
1. Please adhere strictly to the aim of the study in the Introduction section, which is too general in my opinion.
Response: We would like to thank this reviewer for his positive comments. In the revised MS, the introduction section has been modified to adhere strictly to the aim of the study (e.g., Understanding PITX2-dependent atrial fibrillation mechanisms through computational models). In detail, we described the dangers of atrial fibrillation, our limited understanding of PITX2-dependent atrial fibrillation, the advantages of multi-scale computing models and a summary of this narrative review.
Changes in the introduction section (Pages:2-3, lines: 52-77):
The paired-related homeobox gene (PITX2) has been considered to be a potential gene that may trigger AF risk variants on chromosome 4q25[5, 6] and two SNPs (rs2200733 and rs10033464) in chromosome 4q25 were reported[7]. Although various advanced technologies have been developed, their effectiveness is limited and existing treatment regimens are rarely curative[8-10]. This result is partly due to our limited understanding of the mechanisms of AF in the first and second stages (Fig. 1A). In the third stage, the development of actionable personalized approaches, which take into account patient-specific profiles and arrhythmia mechanisms, will likely be essential to overcome current challenges in AF management (Fig. 1B). Computational modelling and simulation are indispensable in cardiac electrophysiology in the study of complex arrhythmias[11-13], such as AF. The multi-scale model of cardiac electrophysiology provides a framework that can integrate experimental and clinical findings[14], and link micro-scale phenomena to emerging behaviors of the entire organ[15]. Computational modeling is now an important part of AF mechanism research, because it can supplement experimental observations and suggest novel mechanisms[16-27]. In addition, the entire atrial simulation is currently used to design novel and personalized treatment strategies, thereby contributing to the development of precision medicine in cardiology. In this narrative review, we focus on the latest developments in the atrial model in elucidating the mechanism of PITX2-dependent AF. We summarize experimental studies of the role of PITX2 in cardiogenesis and arrhythmogenesis, advances in atrial modelling, and modelling studies for investigating PITX2-dependent AF mechanisms. Specifically, we summarize the progress in the development of multi-scale AF models, and then focus on the mechanistic connection between alternations in atrial structure and electrophysiology with PITX2-dependent AF from the perspective of computational modeling. We focus on how AF modeling can supplement experimental data in ways that cannot be achieved outside of the simulation framework, and how AF models can reveal novel AF mechanisms. We conclude the review with a summary of the future prospects of the atrial model's mechanical understanding of AF, towards the goal of understanding patient-specific AF mechanisms that would allow for personalised treatment.
2. Could you please describe the practical implications of the study?
Response: We would like to thank this reviewer for his suggestions. In the revised MS, the clinical relevance of the study has been included in the 4.5 sub-section (Clinical relevance and challenges). In detail, we described changes in PITX2 mRNA in AF patients, the effectiveness of anti-arrhythmic drug therapy in PITX2-dependent AF patients, the limited success of rhythm-control therapy in individual patients, and the role of mechanistic computational models for improving rhythm therapy and using in safety pharmacology.
Changes in the 4.5 sub-section (Pages:17-18, lines: 489-521):
Clinical observational studies have suggested that common SNPs on chromosome 4q25 associated with AF modulates response to anti‐arrhythmic drug therapy in patients[125]. Patients with low levels of PITX2 mRNA and AF have also been shown to have improved effectiveness of Class I anti‐arrhythmic drug therapy[60]. PITX2 levels vary markedly in human atriums[60, 62] and it may be desirable to target AF patients with low PITX2 as a distinct population for therapy. The current limited success of rhythm-control therapy is thought to be in part due to heterogeneity of the underlying substrate, interindividual differences, and our inability to predict response to antiarrhythmic drugs in individual patients[74]. By using mechanistic computational models, these observations of PITX2‐dependent effects may help improve rhythm therapy in the future. At present, the relevance of mechanistic computational models is only indirect, serving as a plausibility check for mechanisms proposed based on experimental observations and helping to generate new hypothesis that can subsequently be tested experimentally[16, 18, 20, 22, 23, 95]. Their recent use in safety pharmacology may further affect clinical practice by guiding the preclinical development of novel antiarrhythmic drugs[17]. Similarly, patient-level cost-effectiveness models may also affect clinical practice by influencing reimbursement policies. The direct clinical application of mechanistic models to guide AF therapy (notably ablation) is emerging, but is currently restricted to a few expert centers.
3.Could you please show the limitations of the study?
Response: Thanks for your suggestions. In the revised MS, numerous challenges of computational models used for understanding PITX2-dependent AF have been summarized in the 4.5 sub-section (Clinical relevance and challenges).
Changes in the 4.5 sub-section (Pages:17-18, lines: 489-521):
Taken together, currently available models have provided insight into all major components of AF therapy, including antiarrhythmic drugs, ablation and anticoagulation, but their role in the disease management of AF patients is still in its infancy and there are numerous challenges.
- Each action potential model has different advantages and disadvantages, with numerous results being model specific.
- The etiology of AF is diverse, but currently available cardiomyocyte models only have limited options for tailoring models to specific clinical conditions.
- Only a handful of labs worldwide have the available expertise, computing power and required collaboration between clinicians, scientists and engineers to apply mechanistic whole-atria models in the clinical setting.
- The extent of personalization of whole-atria models, particularly with regard to electrophysiological properties, remains very limited.
- Current patient-level models do not incorporate fundamental mechanistic patterns of AF pathophysiology.
- Integration of mechanistic modeling with “big data” approaches might help to improve AF diagnosis and management.
4.What is the time frame and type of the included in this review studies?
Response: We searched articles from PubMed and Web of Science database published before 15 June 2021. The searching keywords were “PITX2” and “Atrial fibrillation”. Document type was set to be “Articles”. Dr Jieyun Bai reviewed all relevant articles (168 articles) to collect potential eligible articles.

Round 2
Reviewer 2 Report
Thank you for addressing my queries. I have no further comments.